

# Facilitation of a free-roaming apex predator in working lands: evaluating factors that influence leopard spatial dynamics and prey availability in a South African biodiversity hotspot

Eugene Greyling[1,2,*], Jessica Comley[3,4,*], Michael I. Cherry[1], Alison J. Leslie[5] and Lana Müller[2]

[1] Department of Botany & Zoology, Stellenbosch University, Stellenbosch, Western Cape, South Africa
[2] The Cape Leopard Trust, Cape Town, Western Cape, South Africa
[3] Wildlife and Reserve Management Research Group, University of Mpumalanga, Mbombela, Mpumalanga, South Africa
[4] Current Affiliation: Department of Environmental and Life Sciences, Universiti Brunei Darussalam, Brunei Darussalam
[5] Department of Conservation Ecology & Entomology, Stellenbosch University, Stellenbosch, Western Cape, South Africa
* These authors contributed equally to this work.

Corresponding author
Eugene Greyling,
eugene@capeleopard.org.za

## ABSTRACT

Apex predators ideally require vast intact spaces that support sufficient prey abundances to sustain them. In a developing world, however, it is becoming extremely difficult to maintain large enough areas to facilitate apex predators outside of protected regions. Free-roaming leopards (*Panthera pardus*) are the last remaining apex predator in the Greater Cape Floristic Region, South Africa, and face a multitude of threats attributable to competition for space and resources with humans. Using camera-trap data, we investigated the influence of anthropogenic land modification on leopards and the availability of their natural prey species in two contrasting communities—primarily protected (Cederberg) and agriculturally transformed (Piketberg). Potential prey species composition and diversity were determined, to indicate prey availability in each region. Factors influencing spatial utilisation by leopards and their main prey species were also assessed. Estimated potential prey species richness (Cederberg = 27, Piketberg = 26) and diversity indices (Cederberg—$H' = 2.64$, $Ds = 0.90$; Piketberg—$H' = 2.46$, $Ds = 0.89$), supported by both the Jaccard's Index ($J = 0.73$) and Sørensen's Coefficient ($CC = 0.85$), suggested high levels of similarity across the two regions. Main leopard prey species were present in both regions, but their relative abundances differed. Grey rhebok, klipspringer, and rock hyrax were more abundant in the Cederberg, while Cape grysbok, Cape porcupine, chacma baboon, and common duiker were more abundant in Piketberg. Leopards persisted across the agriculturally transformed landscape despite these differences. Occupancy modelling revealed that the spatial dynamics of leopards differed between the two regions, except for both populations preferring areas further away from human habitation. Overall, anthropogenic factors played a greater role in affecting spatial utilisation by leopards and their main prey species in

the transformed region, whereas environmental factors had a stronger influence in the protected region. We argue that greater utilisation of alternative main prey species to those preferred in the protected region, including livestock, likely facilitates the persistence of leopards in the transformed region, and believe that this has further implications for human-wildlife conflict. Our study provides a baseline understanding of the potential direct and indirect impacts of agricultural landscape transformation on the behaviour of leopards and shows that heavily modified lands have the potential to facilitate mammalian diversity, including apex predators. We iterate that conservation measures for apex predators should be prioritised where they are present on working lands, and encourage the collaborative development of customised, cost-effective, multi-species conflict management approaches that facilitate coexistence.

# INTRODUCTION

Apex predators are generally large carnivores that can act as keystone species and as such, they have been labelled as ecosystem engineers (*Palazón, 2017*). One way in which apex predators primarily influence ecosystems is by exhibiting prey species control: they can directly reduce prey species numbers by predating on them (reducing competitive exclusion among herbivores, thus inducing greater diversity), but also by influencing them indirectly through behavioural changes, which affect ecosystem resources (*Miller et al., 2001*; *Frank, 2008*; *Estes et al., 2011*; *Rosenblatt et al., 2013*). As such, the local extinction of apex predators within ecosystems can often bear drastic trophic cascade consequences (*Terborgh et al., 2001*; *Hebblewhite et al., 2005*; *Ripple et al., 2014*, *2016*; *Suraci et al., 2016*). For example, biodiversity can be reduced (*Estes et al., 2011*) while the transmission of infectious diseases to humans (*Keesing et al., 2010*) and damages to crops can increase (*Brashares et al., 2013*). Apex predators also tend to be seen as charismatic species sought after by tourists and hunters (*Lindsey et al., 2007*; *Van der Meer, Badza & Ndhlovu, 2016*), thereby holding an important economic value to society. Thus, the disappearance of apex predators from ecosystems is likely to stimulate knock-on effects which may adversely impact human wellbeing (*Díaz et al., 2006*; *Estes et al., 2011*).

Traditionally, it was believed that vast, relatively intact ecosystems were required to effectively support viable apex predator populations (*Sillero-Zubiri & Laurenson, 2001*; *Morrison et al., 2007*). Their high trophic position and large body size suggests that they require extensive home ranges which sustain sufficient prey abundances (*Morrison et al., 2007*; *Ripple et al., 2014*). These spatial requirements often bring apex predators into conflict with humans (*Inskip & Zimmerman, 2009*; *Nyhus, 2016*) as they are amongst the first species to be affected by the expansion of human populations and associated cultivation of previously untouched habitats (*Morrison et al., 2007*; *Ripple et al., 2014*; *Aebischer et al., 2020*). A species that is influenced by anthropogenic development to an increasing extent

across the globe is the leopard (*Panthera pardus*; Linnaeus, 1758). Leopards are the most widespread large felid, occurring across much of Africa and tropical Asia (*Nowell & Jackson, 1996*; *Stein et al., 2020*). They are very adaptable and successfully occupy a large variety of habitats across their range (*Jacobson et al., 2016*; *Stein et al., 2020*), including areas alongside large urban spaces (*Kuhn, 2014*; *Braczkowski et al., 2018*). However, their ability to inhabit areas in such close proximity to humans makes them particularly susceptible to competition with humans for space and resources, inevitably placing leopards at great risk.

Globally, leopards are considered as Vulnerable as their populations are declining and they face multiple threats to their survival (*Stein et al., 2020*). An estimated 75% of their historic range has been lost (*Jacobson et al., 2016*), where the average loss for large carnivore species is only around 53% (*Ripple et al., 2014*). Although suitable habitat in southern Africa—arguably hosting the healthiest leopard population across the species' range (*Stein et al., 2020*)—remains widely distributed, it is highly fragmented, having experienced ~51% decline since 1750 (*Jacobson et al., 2016*). Anthropogenic activities, in particular agricultural practices, are primarily deemed responsible for this fragmentation (*Swanepoel et al., 2013*). Indeed, *Brink & Eva (2009)* showed that agricultural land increased by 57% at the expense of natural vegetation in sub-Saharan Africa in just 25 years (1975–2000). In South Africa, ~68% of remaining habitat suitable for leopards is found in areas that are naturally susceptible to land-use transformation (*Swanepoel et al., 2013*). Leopards that occupy these non-protected regions are most at risk of being killed by human-induced causes such as snares, hunts, poison, or motor vehicle collisions (*Balme, Slotow & Hunter, 2010*; *Swanepoel et al., 2013*, *2015*). Consequently, it is vitally important that conservation measures be established to accommodate free-roaming leopards across transformed landscapes to facilitate functional population connectivity and ensure ecosystem resilience (*Balme, Slotow & Hunter, 2010*; *Swanepoel et al., 2013*; *Swanepoel, Somers & Dalerum, 2015*). Most research on leopards in South Africa has taken place inside protected areas (*Balme et al., 2014*), which means that inadequate data is likely jeopardizing the conservation of the species on working lands.

Free-roaming leopards are the last remaining apex predator found in the Cape Floristic and Succulent Karoo Regions (*Martins & Martins, 2006*), which are both biodiversity hotspots and together forms the Greater Cape Floristic Region (*Born, Linder & Desmet, 2007*), in the Western Cape province of South Africa. Here, leopards generally occupy considerably larger home ranges (*Patterson, 2008*) and occur at much lower densities (*Martins & Martins, 2006*) than leopards found elsewhere in Africa (excluding the Kgalagadi; *Mizutani & Jewell, 1998*; *Bothma & Bothma, 2012*). Furthermore, these leopards are also considered to be smaller on average than most leopards (excluding Arabian leopard; *Spalton & Al Hikmani, 2006*) found elsewhere throughout the species' range (*Stuart, 1981*; *Martins & Martins, 2006*). Almost 90% of the total area of the Western Cape is regarded as potential farmland, and the human population of the province (±55 people per km$^2$; *Statistics South Africa, 2021*) has consistently grown at a faster rate than the national average (*Partridge, Morokong & Sibulali, 2021*). The Western Cape is therefore an ideal location to investigate the influence of landscape transformation, both directly and indirectly, on this apex predator.

The aforementioned characteristics of leopards in the Western Cape all presumably reflect adaptation to a different diet (*Martins et al., 2010*). With a great diversity in habitat usage, leopards opportunistically hunt a wide range of prey, depending on local availability (*Hayward et al., 2006*). A decline in their primary prey base can, however, impact leopard population structure (*Marker & Dickman, 2005*; *Ray, Hunter & Zigouris, 2005*; *Wolf & Ripple, 2016*), and may also affect human-wildlife conflict levels by altering leopard behaviour (*Khorozyan et al., 2015*). Human-wildlife conflict has long been prevalent in the Western Cape and remains a complex challenge to this day (*Martins & Martins, 2006*; *Nieman, Wilkinson & Leslie, 2020*). Indeed, leopards within this region are not only deemed responsible for livestock losses, but direct conflict also exists with leopard prey species, often regarded as crop raiders (G Malherbe–Off-reserve Conservation Manager at CapeNature, G Malherbe, 2021, personal communications).

The primary (main) prey base for leopards across the Western Cape include common duiker (duiker; *Sylvicapra grimmia*, Linnaeus, 1758), Cape grysbok (grysbok; *Raphicerus melanotis*, Thunberg, 1811), klipspringer (*Oreotragus oreotragus*, Zimmermann, 1783), and grey rhebok (rhebok; *Pelea capreolus*, Forster, 1790), as well as rock hyrax (hyrax; *Procavia capensis*, Pallas, 1766), Cape porcupine (porcupine; *Hystrix africaeaustralis*, Peters, 1852), and chacma baboons (baboon; *Papio ursinus*, Kerr, 1792) (*Martins et al., 2010*; *Drouilly, Nattrass & O'Riain, 2018*; *Mann et al., 2019*; *Müller et al., 2022a*). In addition to suffering retaliatory killings, these species are also targeted for the illegal harvesting of bushmeat by means of snaring in the province (*Nieman et al., 2019*). An akin competitive relationship between humans and leopards has previously been documented in the Congo Basin (*Henschel et al., 2011*), with profound negative consequences for leopards. Illegal hunting, which impacts prey species availability, is a continuous concern for wildlife worldwide (*Lindsey et al., 2013*; *Heurich et al., 2018*). Besides, snares can also be responsible for the direct capture of leopards resulting in severe injury or death (*Swanepoel et al., 2015*; *Williams et al., 2017*; *Nieman, Leslie & Wilkinson, 2019*; *Gubbi, Kolekar & Kumara, 2021*).

As the only terrestrial apex predator to persist in the Western Cape, free-roaming leopards are expected to be sensitive to changes in prey species populations. Yet, little is known about the extent to which this relationship may be influenced by humans in the context of commercial agriculture. Several studies investigating the drivers of leopard occurrence, density, or ranging behaviour (*e.g. Jiang et al., 2015*; *Allen et al., 2020*; *Searle et al., 2020*; *Snider et al., 2021*; *Loveridge et al., 2022*), and predator-prey overlap (*e.g. Dias, de Campos & Rodrigues, 2018*; *Havmøller et al., 2020*; *Palei et al., 2022*; *Sehgal et al., 2022*; *Zaman et al., 2022*), have been performed worldwide. However, until relatively recently, few studies have investigated African leopard (*P. p. pardus*) ecology in human-disturbed landscapes (see *Marker & Dickman, 2005*; *Williams et al., 2017*; *Strampelli et al., 2018*)—particularly in commercial agricultural regions. Furthermore, limited comparisons of predator populations between analogous protected and non-protected regions have been made (*e.g. Swanepoel, Somers & Dalerum, 2015*; *Drouilly, Nattrass & O'Riain, 2018*; *Curveira-Santos et al., 2020*; *Faure et al., 2021*). Considering that half of all habitable land worldwide is used for agriculture (*Ellis et al., 2010*; *Ritchie & Roser, 2013*)—regarded as the

biggest driver of terrestrial habitat loss (*IPBES, 2019*)—our understanding of factors which could best enable (or inhibit) the persistence of apex predators in agriculturally transformed environments, including variables influencing the availability of their natural prey species, is vital to aid management decision making and account for in cost-benefit models that aim to minimise conflict. Inferences about spatial variation in species composition and diversity (richness and evenness; *Colwell, 2009*) are also important, both to ecological hypotheses about structure and function of communities and to community-level conservation management (*Nichols et al., 1998*).

This study investigated potential prey species diversity, community structure, and factors affecting the spatial behaviour of leopards and their main prey species, in both an agriculturally transformed and a largely protected landscape. First, we aimed to determine whether any real differences exist with reference to potential prey species found in each community and whether any such difference is evidently reflected in the respective leopard subpopulations. Second, we aimed to evaluate and compare drivers of habitat utilisation by leopards and their main prey in each region. We anticipated a greater diversity of potential leopard prey species to exist in the protected community, as well as differences in community structure. Relative leopard and prey abundances were predicted to be lower in the transformed agricultural region, and anthropogenic factors were expected to negatively influence space-use, by leopards as well as their main prey species, across the greater landscape.

## MATERIALS AND METHODS

### Ethical statement

Relevant permissions to conduct our research were granted by the Social, Behavioural and Education Research Ethics Committee at Stellenbosch University (Project ID #15315), CapeNature (Permit #CN44-59-12321), and in writing by all landowners involved. Data collection was performed using camera-traps, which is a non-invasive research method. All data, including any images captured of human subjects, were treated as strictly confidential.

### Study area

The Piketberg region (hereafter Piketberg) encompasses a transformed landscape that is primarily characterised by mixed agricultural practices. Our study area is located approximately 130 to 160 km north of Cape Town, South Africa, and situated southwest of the Cederberg region (Fig. 1). The survey area was approximately 1,500 km$^2$ in size (53–864 m above sea level), covering 55 privately owned mixed agricultural farms extending north from the town of Piketberg to Paleisheuwel, with Citrusdal in the east and Aurora on the western boundary (Fig. 1). The area consists of natural vegetation forming a mosaic, highly fragmented by livestock (*e.g.* cattle (*Bos taurus*, Linnaeus, 1758), horses (*Equus ferus caballus*, Linnaeus, 1758), sheep (*Ovis aries*, Linnaeus, 1758), goats (*Capra hircus*, Linnaeus, 1758), pigs (*Sus domesticus*, Erxleben, 1777)), fruit, and other crop farmlands in and around mountainous terrain (*Linder, 1976*; *Mucina & Rutherford, 2006*).
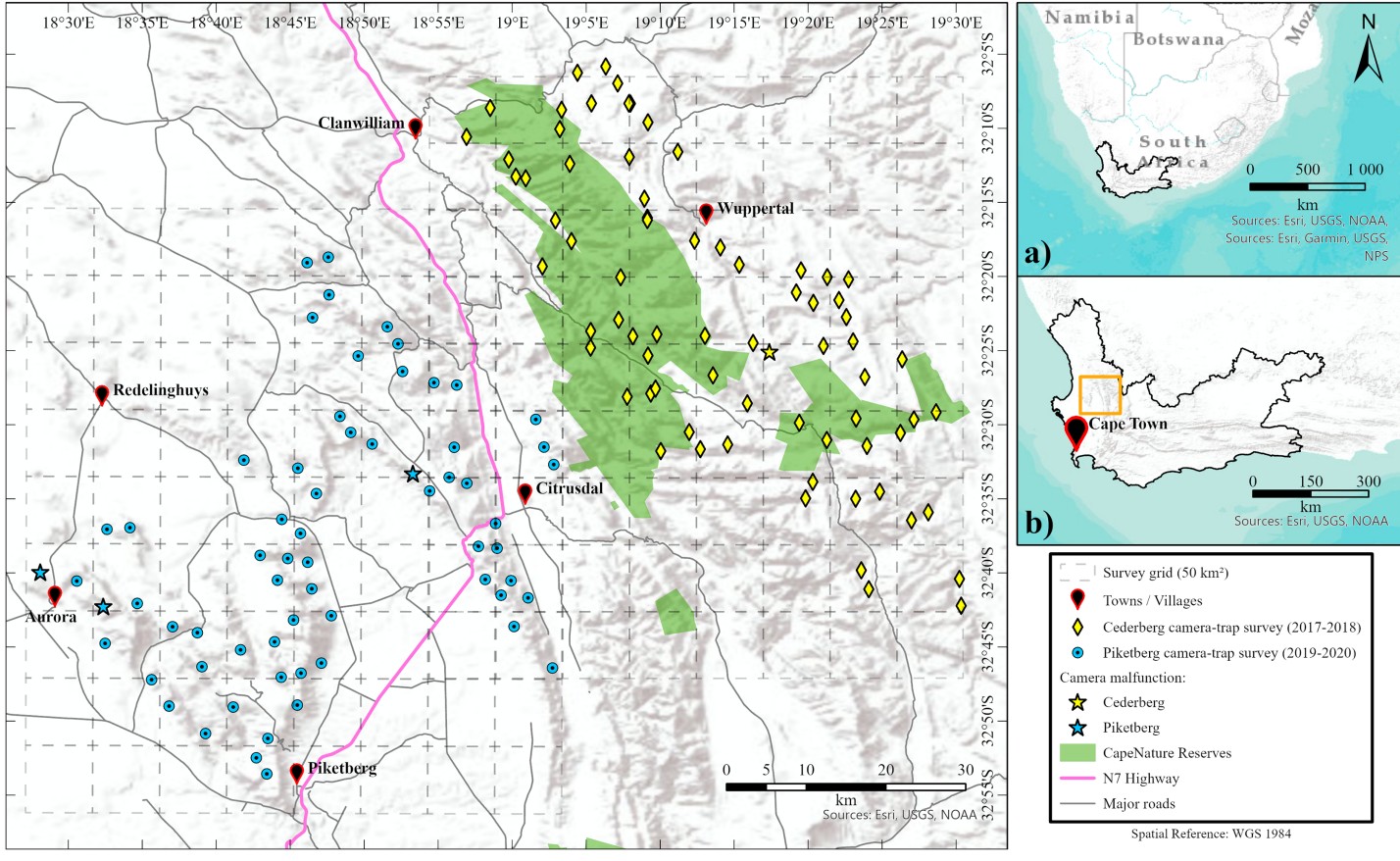

**Figure 1** **Location of survey regions.** Maps showing the location of the Western Cape province within South Africa (A), and the proximity of the survey areas within the Western Cape (B). Main map shows the location of camera-trap stations for both Cederberg and Piketberg. Failed camera-trap stations (because of major camera malfunctioning, fire, other damage, or theft) are noted. Formally proclaimed protected areas (*i.e.*, CapeNature reserves) are also highlighted.

The Cederberg region (hereafter Cederberg), known for its rugged remoteness, is a largely protected area first proclaimed in 1973 and located 200 to 250 km northeast of Cape Town (Fig. 1). This survey area was approximately 1,700 km$^2$ in size (254–1,455 m above sea level) and included the Matjiesrivier Nature Reserve and the Cederberg Wilderness Area. Both areas are formally protected and managed by the provincial conservation regulation body and authority in the Western Cape, known as CapeNature (Fig. 1). Privately owned areas included were Bushmans Kloof Wilderness Reserve, community owned land used for the harvesting of rooibos and limited pastoralism by subsistence farmers, and the Cederberg Conservancy, consisting of pro-conservation farms that are used for ecotourism and largely kept in a natural ecological state. The two main biomes present are Fynbos and Succulent Karoo in mountainous terrain (*Mucina & Rutherford, 2006*).

The Western Cape has a Mediterranean type climate characterised by hot, dry summers and cold, wet winters (*Cowling & Holmes, 1992*). Localised climatic conditions vary across the greater study area because of its mountainous nature, and the vastness of the landscape makes it extremely difficult to describe the climate of the study area in general terms.

In essence, average annual rainfall appears slightly higher, and average summer air temperatures somewhat lower in Piketberg compared to the Cederberg (*Climate-Data.org, 2020*). Average winter air temperatures are more uniform across both regions, but snowfall is more abundant in the Cederberg which is generally at a higher altitude (*Climate-Data.org, 2020*).

## Field sampling methods

### Camera-trap surveys

Single season (dry summer—November to March) subsets of photographic data that were collected in the Cederberg (2017–2018) and Piketberg (2019–2020) were used for our study. The 73 camera-trap stations (*n* = 146 cameras) in the Cederberg were all located within areas with protected status, whereas the 64 camera-trap stations (*n* = 128 cameras) in Piketberg were situated in non-protected areas nestled between and on farmlands (Fig. 1). Paired motion and heat detecting Cuddeback X-Change series camera-traps were used at each camera station during both surveys. Our setup procedures followed standard protocols optimised for the detection of leopards, whereby the landscape across both study regions was divided into 50 km$^2$ blocks (Fig. 1), based on the minimum estimated home range size recorded for a female leopard with cubs in the Western Cape (37 km$^2$; *Martins, 2010*; also see *Müller et al., 2022a*, *2022b*). Camera-trap stations were dispersed across mountainous habitat with mean distances of 2.78 km (Cederberg) and 3.09 km (Piketberg) between individual stations and two to three stations per block (Fig. 1). Camera-trap locations were selected based on the presence of tracks and signs of leopards and their main natural prey species found in the nearby vicinity on initial site investigation. Cameras were mounted ~40 cm above the ground and perpendicular to a game trail, road, or drainage line. Camera-traps were serviced at an interval of roughly 8 weeks to download images, change batteries, perform site data collection (*i.e.*, recording covariates around each site), and for general maintenance.

### Covariate data

Environmental (*i.e.*, altitude, vegetation type, vegetation age, nearest water source type and its distance) and anthropogenic (*i.e.*, distances to the nearest road and human habitation, evidence of disturbance, livestock, and/or hunting) covariates surrounding each camera-trap station were recorded during each servicing period (see Supplementary Material, Table S1). Some categorical and binomial covariates were determined with a repeated physical site inspection of the surroundings performed by a trained and experienced individual covering a 100 m radius around each station (Table S1). Other variables were determined by a combination of physical investigation, with recordings made using a GPS unit (ETrex 10; Garmin, Olathe, KS, USA), and local knowledge (Table S1). Any further verification was performed using Google Earth (*Google, 2019*) and historical fire records (Table S1). The chosen covariates reflect natural and human-induced factors that could influence predator and prey space-use or detection at each site.

## Data analyses

### Camera-trap data

Camera Base software (*Tobler, 2010*) was used to process images and extract meta-image information from each photograph (image name, date, and time) while correcting for any time stamp errors. Faunal species and number of individuals in each photograph was identified where possible. Primary analyses were performed using the Camera Trap Analysis Package (CTAP) software developed by the Zoological Society of London (*Amin & Wacher, 2017*). Only terrestrial mammals >0.5 kg, including leopard (see *Charsley (1977)*, *Steyn & Funston (2006)*, and *Balme & Hunter (2013)* for examples of reported cannibalism), were considered as potential leopard prey species for analyses. They are the main target group for camera-traps set up in this manner and are also generally regarded as the main dietary component of leopards (*Hayward et al., 2006*; *Tobler et al., 2008*; *Martins et al., 2010*; *Drouilly, Nattrass & O'Riain, 2018*; *Mann et al., 2019*; *Müller et al., 2022a*). Relative abundance indices (RAI; *i.e.,* number of events, where an "event" is defined as any image sequence for a given species occurring after an interval of ≥60 min from a previous sequence of that species, per 100 days of camera trapping; *Karanth & Nichols, 1998*; *Amin et al., 2018*) per camera-trap station were calculated for known main prey species of leopards in the study area, sympatric meso-carnivores—caracal (*Caracal caracal*, Schreber, 1776) and black-backed jackal (*Canis mesomelas*, Schreber, 1775)—and leopards, and used as biotic covariates during occupancy modelling (Table S1). Despite being influenced by sampling design or species' behaviour (*Sollmann et al., 2013*), RAI is still considered a suitable tool for assessing species occurrence (*Hedwig et al., 2018*; *Palmer et al., 2018*).

### Community structure

Biological communities can differ in species composition, total number of species (richness), and the relative abundance of species (evenness) (*Colwell, 2009*). Species sample-based rarefaction curves were constructed and the terrestrial medium-to-large (>0.5 kg) mammal species richness ($S$), representing potential leopard prey species, was estimated for each surveyed community using a non-parametric incidence-based estimator Jackknife with order one (*Bunge & Fitzpatrick, 1993*). Livestock and other domestic species were excluded from analyses. We calculated Simpson's ($Ds$) and Shannon-Wiener ($H'$) diversity indices for each community using global RAI values in the package 'vegan' in R statistical software (see Table S2; *Oksanen et al., 2019*). Simpson's diversity index is most sensitive to changes in more common highly abundant species, while the Shannon-Wiener diversity index is most sensitive to changes in rare less abundant species (*Magurran, 2004*). Community structure plots representing the RAI as a factor of trophic level and mean adult body weight of potential prey species were also constructed. Jaccard's Index ($J$) and the Sørensen's Coefficient ($CC$) were calculated as measures of similarity, directly comparing Piketberg and the Cederberg, using the following formulae:

$$J = \frac{A}{[A + B + C]} \qquad\qquad CC = \frac{2A}{[2A + B + C]}$$

$A$ = Number of species shared by two communities,

$B$ and $C$ = Number of species unique to each of the two communities, respectively.

The latter places more emphasis on the shared species present rather than the unshared and retains sensitivity in more heterogeneous data sets. Sørensen's ecological distance ($D_{CC} = 1 - CC$) is therefore useful as many species may potentially be present in a community, but not present in a sample from that community (*Magurran, 2004*).

### Occupancy modelling

Naïve occupancy, defined as the proportion of sites that recorded at least one photograph of the target species, was calculated for leopards of each community:

$$\psi_{na\bar{i}ve} = \frac{\#\ of\ sites\ detected}{\#\ of\ sites\ sampled}$$

To accurately model occupancy, unique detection histories consisting of 1s (detection) and 0s (non-detection) were created for leopards and their main prey species (*i.e.*, baboon, duiker, grysbok, hyrax, klipspringer, porcupine, and rhebok) in the Cederberg and Piketberg. The unique detection histories reflected the presence or absence of each species at each camera-trap site on each occasion (maximum value '1' per 24 h period) for each region. Original unique detection history datasets (Cederberg N occasions = 151; Piketberg N occasions = 132) for each species were collapsed into data subsets by merging the occasions into intervals of between five to 11-day sampling occasions. This was deemed appropriate as it reduced each species dataset into manageable sizes for computational purposes and accurately represented the rarity of the study species (*Sollmann, 2018*).

All continuous covariate values were scaled into standardized z-scores (*Bruggeman et al., 2016*). Multi-collinearity was tested for by calculating variance inflation factors (VIF), whereby covariates with VIF scores greater than three were removed (*Wang et al., 2018*). A global occupancy model that included all ecologically relevant covariates (see Table S1) was applied to the subsets of data for each species and tested for goodness-of-fit (*MacKenzie & Bailey, 2004*). Subset data for each species that had the closest over dispersion statistic ($\hat{c}$) to 1 (extreme values over (>3) or under 1 (<0.90) indicate poor fit of the data) and an insignificant chi-square probability ($\chi^2 p > 0.05$) was chosen for further occupancy analyses (see Table S3; *Mazerolle, 2017*). This showed maximum model fit without over compressing statistical power of the data (*Burnham & Anderson, 2004*; *MacKenzie & Bailey, 2004*). Our study violates the assumption of spatial autocorrelation and independence of camera-trap sites, which means that our results should be interpreted within the context of area used (*i.e.*, space-use) rather than area occupied (*MacKenzie & Nichols, 2004*).

For each species, only combinations of covariates that could affect the two modelling parameters (space-use probability, $\psi$, and detection probability, $p$) and that presented ecologically reasonable hypotheses were included (*McDonald et al., 2016*). With such a

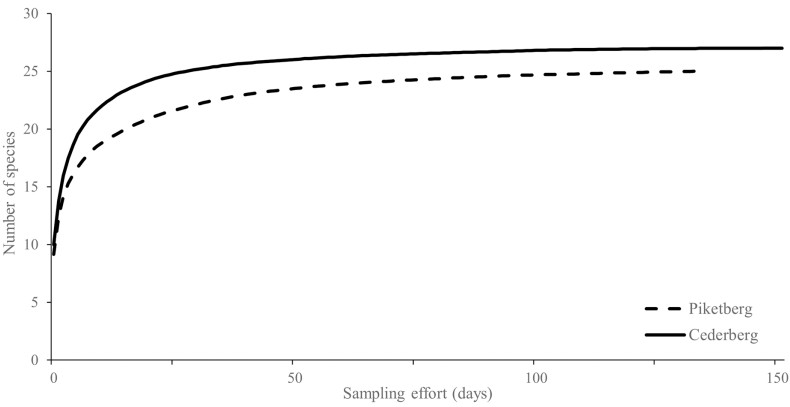

**Figure 2 Rarefied species accumulation curves for medium-to-large (>0.5 kg) terrestrial mammals (*i.e.,* potential leopard prey species) photographed in the Cederberg and Piketberg regions.** Both curves approach an asymptote, indicating sufficient sampling effort.

large number of covariates, the set of candidate models that we might have examined was extremely vast (*Schuette et al., 2013*). Therefore, we used a stepwise procedure following *Dugger, Anthony & Andrews (2011)*, whereby the first step was to model $p$ by investigating additive combinations of covariates while treating $\psi$ as constant (*i.e.*, intercept only). For model selection, the over dispersion statistic ($\hat{c}$) estimated from the global model for each species was used to compute quasi-likelihood information criteria (QAICc: for small sample sizes) by scaling the log-likelihood of each model, for each species, by its corresponding $\hat{c}$ value (*Mazerolle, 2017*). Therefore, QAICc model-selections were used to retain the best $p$ model for each species to use in subsequent analyses of factors affecting $\psi$. The second step was to model $\psi$ by investigating additive combinations of covariates. The package 'unmarked' (*Fiske & Chandler, 2011*) was used to fit models and to estimate covariate coefficients for each parameter in R (version 4.1.2, *R Development Core Team, 2017*). The R package 'AICcmodavg' was used for all model selection computations (*Mazerolle, 2017*).

The lowest ΔQAICc scores (<2) and highest QAICc weights ($w$ >0.10) were used to select the best-approximating models for each species in each community (see Tables S4 and S6; *Burnham & Anderson, 2004*). We drew conclusions about strength of evidence of relationships between covariates and parameters based on 95% confidence intervals (CIs) of coefficients and the direction of relationships (see Tables S5 and S7; *Arnold, 2010*).

## RESULTS

A total of 10,114 operational camera-trap days (mean 140 days/station) were accumulated in the Cederberg, with only one station failure (refer to Fig. 1). In Piketberg, 6,258 operational camera-trap days (mean 103 days/station) were accumulated, and three camera-trap stations failed (Fig. 1). The outcomes of our study are unlikely to be affected by this difference as sampling effort across both regions was sufficient (see Fig. 2) and indices based on relative abundance values were employed, providing equal weight to both communities, therefore allowing comparison.

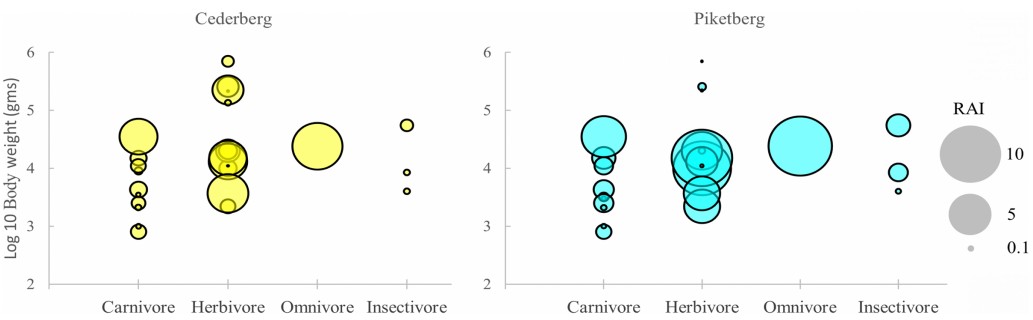

**Figure 3 Distribution of medium-to-large (>0.5 kg) terrestrial mammals (*i.e.*, potential leopard prey species) in the Cederberg and Piketberg on the basis of body size and trophic category.** Each circle represents a species in functional space. The size of the circle is proportional to the trapping rate (RAI) for that species.                                                

## Species richness

Piketberg had an estimated potential prey species richness ($S$ = 26) similar to that of the Cederberg ($S$ = 27). The rarefied species accumulation curves do however show more species detected per unit effort in the Cederberg compared to Piketberg (Fig. 2). Both the Shannon-Wiener ($H'$) and Simpson diversity ($Ds$) indices were only marginally higher in the Cederberg ($H'$ = 2.64, $Ds$ = 0.90) compared to Piketberg ($H'$ = 2.46, $Ds$ = 0.89). Thirty potential natural prey species were photographed across the two regions (Cederberg: 27; Piketberg: 25), with five species unique to the Cederberg and three to Piketberg (Table S2).

## Community structure

Differences in community structure were largely complementary, providing a similar pattern across trophic guilds for both the Cederberg and Piketberg (Fig. 3). The same number of carnivore species were detected in both regions ($n$ = 10), but no Cape fox (*Vulpes chama*, Smith, 1833) or large grey mongoose (*Herpestes ichneumon*, Linnaeus, 1758) were recorded in the Cederberg, whereas black-backed jackal and water mongoose (*Atilax paludinosus*, Cuvier, 1826) were not detected in Piketberg. Herbivores were the most frequently encountered guild across both regions; more herbivorous species were encountered in the Cederberg ($n$ = 13) than in Piketberg ($n$ = 11). Main leopard prey species (*i.e.*, baboon, duiker, grysbok, hyrax, klipspringer, porcupine, and rhebok) were all accounted for in both communities (Table S2).

Across all guilds, global trap rates (RAIs) for species detected were generally higher in Piketberg (Fig. 3 & Table S2). Noteworthy exceptions included hyrax (Cederberg: 4.05; Piketberg: 3.16), klipspringer (Cederberg: 3.56; Piketberg: 2.41), and rhebok (Cederberg: 0.84; Piketberg: 0.14). A marked difference was the higher RAIs of all carnivores >4 kg—including leopard (Cederberg: 3.52; Piketberg: 4.65)—as well as most medium (>0.5 kg and <100 kg) herbivores, in the Piketberg region (Fig. 3 & Table S2). Baboons were the most frequently detected species across both communities (Cederberg: 608 events; Piketberg: 592 events), followed by hyrax in the Cederberg (410 events) and porcupine in Piketberg (540 events). No single species dominated (*i.e.*, >50% of total trap rates) in any of the two communities.

High levels of similarity between the two study regions were shown by both the Jaccard's Index ($J = 0.73$) and Sørensen's Coefficient ($CC = 0.85$). The Jaccard's Index suggested that 73% similarity exists between the contrasting communities. Similarly, the Sørensen Coefficient suggested that the ecological distance that separate the two regions is merely 15%, supporting close relatedness of ecological make-up in each community.

## Spatial behaviour

Camera-trap stations were excluded from occupancy (*i.e.*, space-use) analyses when cameras were operational for <80% of occasions (see Fig. 1). Baboon and rhebok were excluded due to insufficient model fit (Table S3). Only strong relationships with space-use and detection probabilities are discussed (see Table 1). In essence, factors that strongly influenced the spatial dynamics of leopards and their main prey species in the Cederberg were primarily environmental (56%), whereas anthropogenic (32%) and biotic variables (12%) played a less significant role (Table 1). In contrast, anthropogenic variables (41%) dominated in Piketberg relative to notable influences by biotic (36%) and environmental (23%) factors (Table 1).

### Leopard

Leopards occurred across the landscape in both communities and were detected at 55 camera-trap stations in Piketberg and 60 stations in the Cederberg, providing naïve occupancy estimates of 0.852 (Piketberg) and 0.833 (Cederberg) respectively. The spatial dynamics of leopards differed between the Cederberg and Piketberg, except for both leopard populations preferring to utilise areas further away from human habitations (Table 1; Figs. 4A and 5A). In the Cederberg, leopards also preferred areas in closer proximity to permanent rather than seasonal water sources (Table 1; Fig. 4B) but were more likely to be detected further away from water (Table 1; Fig. 4E). Furthermore, leopards of the Cederberg were generally less likely to be detected in areas where sympatric meso-carnivores were more abundant (Table 1; Fig. 4C) and at higher altitudes (Table 1; Fig. 4D). In Piketberg, leopards were less likely to be detected if present at sites characterised by Sandveld vegetation and in areas utilised by livestock (Table 1; Figs. 5B–5E).

### Main prey species

The space-use of grysbok, hyrax, klipspringer, and porcupine were influenced by slightly different factors in the primarily protected Cederberg and agriculturally transformed Piketberg (Table 1). None of the variables we considered were found to strongly influence duiker space-use (Table 1). In the Cederberg, grysbok preferred areas farther away from roads, porcupine preferred areas of older vegetation, while hyrax preferred areas characterised by Karoo vegetation and seemingly avoided areas where caracal and black-backed jackal were abundant. In Piketberg, both hyrax and klipspringer avoided areas where caracal were more prevalent (black-backed jackal are absent), but porcupine appeared to prefer areas that had greater caracal presence.

The detection probabilities (*i.e.*, detectability) of these prey species were affected by various combinations of factors across the greater landscape (Table 1). In both study

**Table 1 QAICc weights (*w*) for covariates from well-supported models (*w* > 0.10 and ΔQAICc < 2) for each species (leopard + main prey) in each community.** Either the direction of the relationship for covariates from the best fit models or the parameter probability estimate (± standard deviation) for top models are indicated in parentheses.

| Species | Parameter | Covariate | Cederberg | Piketberg |
|---------|-----------|-----------|-----------|-----------|
| **Leopard** | Ψ | Habitation | 0.49 (+)[*] | 0.12 (+)[*] |
| | | Water source (seasonal) | 0.49 (−)[*] | 0.14 (+)[a] |
| | | Prey | 0.49 (−)[a] | |
| | | Livestock (yes) | 0.28 (−)[a] | 0.14 (+)[b] |
| | | Altitude | | 0.26 (−)[a] |
| | | Null | | 0.33 (0.92 ± 0.04) |
| | | | | |
| | *p* | Carnivores | 0.49 (−)[*] | |
| | | Altitude | 0.49 (−)[*] | |
| | | Water | 0.49 (+)[*] | |
| | | Prey | 0.49 (+)[a] | |
| | | Vegetation (Renoster) | | 0.33 (−)[a] |
| | | Vegetation (Riverine thicket) | | 0.33 (+)[b] |
| | | Vegetation (Sandveld) | | 0.33 (−)[*] |
| | | Livestock (yes) | | 0.33 (−)[*] |
| | | | | |
| **Duiker** | Ψ | Road | 0.72 (−)[a] | |
| | | Altitude | 0.72 (+)[a] | |
| | | Carnivores | 0.28 (+)[a] | |
| | | Water source (seasonal) | 0.28 (−)[a] | |
| | | Caracal | | 0.17 (+)[a] |
| | | Water | | 0.13 (+)[a] |
| | | Disturbance (yes) | | 0.10 (+)[a] |
| | | Vegetation (Renoster) | | 0.10 (−)[a] |
| | | Vegetation (Riverine thicket) | | 0.10 (−)[a] |
| | | Vegetation (Sandveld) | | 0.10 (+)[a] |
| | | Null | | 0.22 (0.55 ± 0.08) |
| | | | | |
| | *p* | Altitude | 0.72 (−)[*] | 0.22 (−)[*] |
| | | Leopard | 0.72 (+)[a] | 0.22 (−)[*] |
| | | Water | 0.72 (+)[*] | |
| | | Vegetation age | 0.72 (−)[*] | |
| | | Vegetation (Karoo) | 0.72 (−)[a] | |
| | | Habitation | 0.72 (−)[*] | 0.22 (−)[*] |
| | | Road | 0.72 (−)[a] | 0.22 (−)[*] |
| | | | | |
| **Grysbok** | Ψ | Road | 0.16 (+)[*] | |
| | | Leopard | 0.16 (+)[a] | 0.18 (+)[a] |
| | | Water source (seasonal) | 0.16 (−)[a] | |

(Continued)

| Table 1 (continued) | | | | |
|---|---|---|---|---|
| Species | Parameter | Covariate | Cederberg | Piketberg |
| | | Altitude | | 0.37 (+)[a] |
| | | Disturbance (yes) | | 0.15 (−)[a] |
| | | Vegetation (Karoo) | 0.11 (+)[a] | |
| | | Null | | 0.30 |
| | | | | |
| | p | Vegetation (Karoo) | 0.16 (−)* | |
| | | Road | 0.16 (−)* | 0.37 (−)* |
| | | Disturbance (yes) | 0.16 (+)* | |
| | | Leopard | | 0.37 (+)* |
| | | Altitude | | 0.37 (+)* |
| | | Habitation | | 0.37 (−)* |
| | | | | |
| Hyrax | Ψ | Vegetation (Karoo) | 0.36 (+)* | |
| | | Carnivores | 0.36 (−)* | |
| | | Water | 0.27 (+)[a] | |
| | | Leopard | 0.22 (+)[a] | 0.24 (+)[a] |
| | | Disturbance (yes) | 0.15 (−)[a] | |
| | | Caracal | | 0.37 (−)* |
| | | Habitation | | 0.24 (+)[a] |
| | | | | |
| | p | Vegetation (Karoo) | 0.36 (+)* | |
| | | Road | 0.36 (+)* | |
| | | Altitude | 0.36 (−)* | |
| | | Carnivores | 0.36 (−)[a] | |
| | | Leopard | | 0.37 (−)* |
| | | Caracal | | 0.37 (+)* |
| | | | | |
| Klipspringer | Ψ | Altitude | 0.32 (+)[a] | |
| | | Water source (seasonal) | 0.32 (+)[a] | |
| | | Livestock (yes) | 0.15 (+)[a] | |
| | | Caracal | | 0.17 (−)* |
| | | Water | | 0.12 (+)[a] |
| | | Disturbance | | |
| | | Leopard | | 0.17 (−)[a] |
| | | | | |
| | p | Vegetation (Karoo) | 0.32 (+)* | |
| | | Carnivores | 0.32 (−)* | |
| | | Disturbance (yes) | 0.32 (−)* | |
| | | Water source (seasonal) | | 0.17 (−)* |
| | | Water | | 0.17 (−)* |
| | | Road | | 0.17 (−)* |
| Species | Parameter | Covariate | Cederberg | Piketberg |
|---------|-----------|-----------|-----------|-----------|
| | | | **Table 1 (continued)** | |
| | | Caracal | | 0.17 (−)* |
| **Porcupine** | Ψ | Caracal | | 0.38 (+)* |
| | | Road | | 0.31 (−)[a] |
| | | Disturbance (yes) | | 0.30 (+)[a] |
| | | Vegetation age | 0.38 (+)* | |
| | | Water | 0.38 (+)[a] | |
| | | Leopard | 0.18 (−)[a] | |
| | p | Habitation | | 0.38 (+)* |
| | | Disturbance (yes) | | 0.38 (+)* |
| | | Altitude | | 0.38 (+)[a] |
| | | Vegetation (Karoo) | 0.38 (−)* | |
| | | Vegetation age | 0.38 (−)* | |
| | | Road | 0.38 (−)* | |
| | | Carnivores | 0.38 (+)[a] | |

**Notes:**
[*] Strong evidence of relationship (CI estimates do not overlap 0).
[a] Medium evidence of relationship (CI estimates overlap 0, but are not centred on 0).
[b] Weak evidence of relationship (CI estimates overlap 0 and are centred on 0).

regions, duiker were more likely to be detected (if present) nearer to human habitations and at lower altitudes, grysbok were more likely to be detected in closer proximity to roads, and klipspringer were more likely to be detected where meso-carnivores were less abundant. Considering only the Cederberg, detectability of duiker was greater farther away from water sources and lower in areas consisting of older vegetation. Detection of grysbok was more likely in areas that showed signs of anthropogenic disturbance, but less likely at sites characterised by Karoo vegetation. Similarly, the probability of detecting porcupine was also lower in areas of Karoo as well as older vegetation, and greater in areas closer to roads. By contrast, the detectability of both hyrax and klipspringer in the Cederberg was greater within the Karoo biome. Furthermore, hyrax were more likely to be detected at lower altitudes and farther away from roads, whereas the probability of detecting klipspringer was less at anthropogenically disturbed sites. In Piketberg, detection probability of klipspringer was greater closer to water, particularly permanent water sources. Both klipspringer and duiker were also more likely to be detected in the vicinity of roads in this region. Additionally, duiker, as well as hyrax, were less likely to be detected where leopard RAI was greater. Instead, hyrax detectability increased as caracal RAI increased. On the other hand, grysbok in Piketberg were more likely to be detected in areas where leopards were seemingly more abundant. Their detection probability was also greater at higher altitudes and in areas closer to human habitations. Detectability of porcupine, however, was lower nearby human habitations. Yet, porcupine were more likely

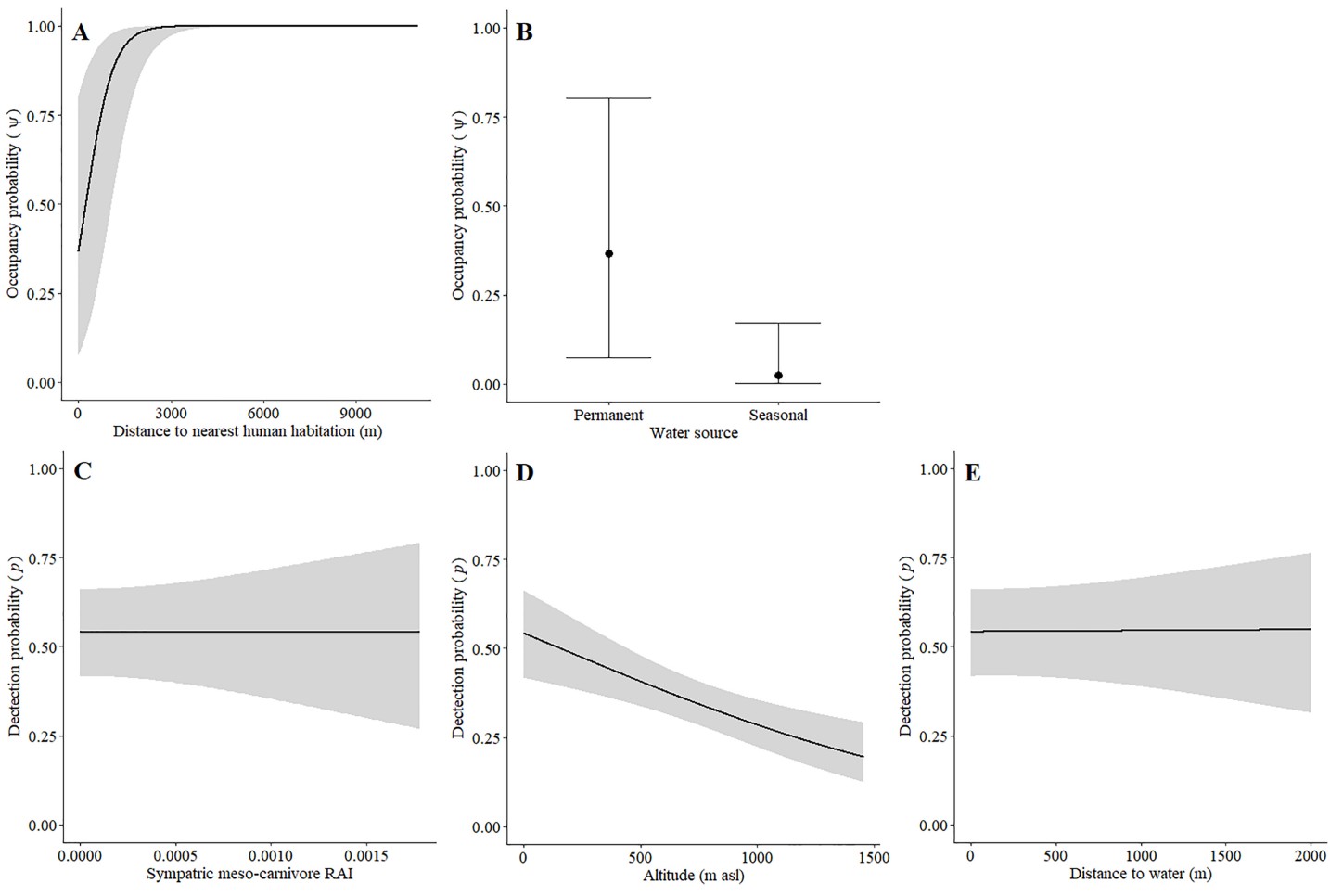

**Figure 4 Detection and space-use (*i.e.*, occupancy) probabilities for leopards with regards to variables with strong associations in the Cederberg.**

to be detected in areas that showed other signs of anthropogenic disturbance across the agriculturally transformed region.

## DISCUSSION

### Prevalence of leopards

Contrary to expectations, our results, in terms of both relative abundance indices (RAI: Cederberg: 3.52; Piketberg: 4.65) and naïve occupancy ($\psi_{naïve}$: Cederberg: 0.833; Piketberg: 0.852), suggest that leopards are at least as relatively widespread across the agriculturally transformed Piketberg landscape, and potentially even more abundant, than they are in the primarily protected Cederberg region. We similarly observed greater relative abundances of all carnivorous species >4 kg shared between the two communities (Fig. 3 & Table S2). Assuming comparable average activity levels, leopards of Piketberg may therefore occupy smaller or more overlapping home ranges than leopards found in the Cederberg (refer to *Müller et al., 2022b*). Recently, *Snider et al. (2021)* showed that it is indeed common for free-roaming leopard home-range size to be smaller (inferring greater density) within areas

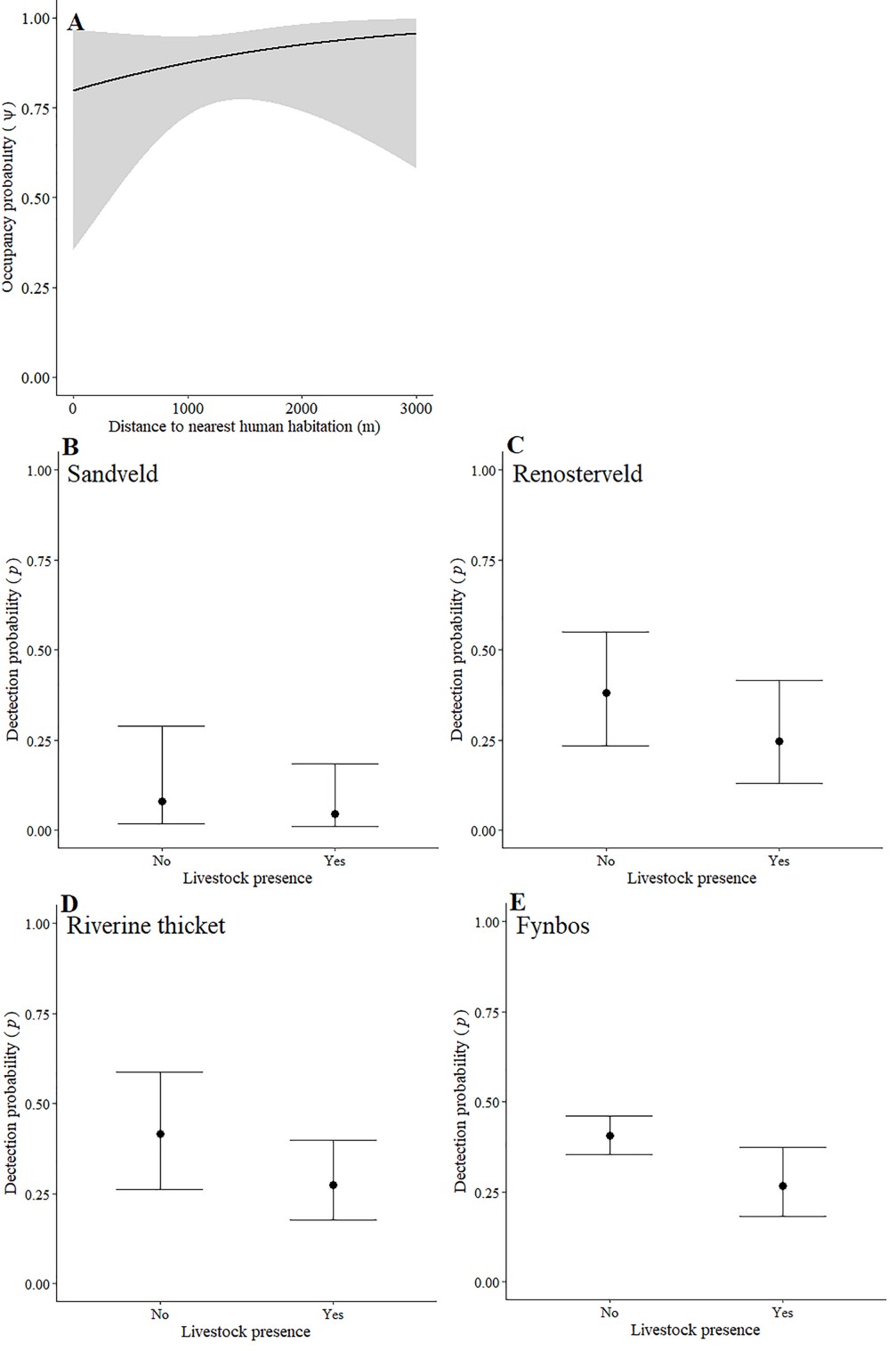

**Figure 5 Detection and space-use (*i.e.,* occupancy) probabilities for leopards with regards to variables with strong associations in Piketberg.**

of greater human population density. Alternatively, should activity levels differ greatly between the two subpopulations, leopards could also occupy larger home ranges in Piketberg (*Neilson et al., 2018*; *Rogan et al., 2019*). Since the relationship between occupancy (or space-use), abundance, and density is influenced by the number of individuals, home range size, and the degree of spatial overlap between individuals, neither relative abundance nor the space-use parameter can be used to ascertain differences in density with absolute certainty (*Rogan et al., 2019*). Nonetheless, greater predator abundances can only be sustained by a greater relative carrying capacity (*i.e.*, prey availability). Higher RAIs were also noted in Piketberg compared to the Cederberg for most medium sized (>0.5 kg and <100 kg) herbivores (Table S2), which are regarded as the primary prey component for leopards (*Hayward et al., 2006*). Therefore, prey populations in Piketberg appeared sufficient to facilitate and sustain relatively greater, potentially denser, and at least as equally successful predator populations to those found in the Cederberg. Notable differences that we observed in the RAIs of main leopard prey species between the two study regions, however, suggests that leopard diet composition is likely to differ (based on availability; *Hayward et al., 2006*) in the agriculturally transformed landscape.

The main prey species considered in this study are known to comprise approximately 85% of the biomass consumed by leopards in the Cederberg (*Müller et al., 2022a*), while in other areas of the Western Cape they comprise as much as 91% (Boland) and as little as 57% (Little Karoo) of leopard diets (*Mann et al., 2019*). Grey rhebok, klipspringer, and rock hyrax were more abundant in the Cederberg, while Cape grysbok, Cape porcupine, chacma baboon, and common duiker were more abundant in Piketberg (Table S2). Leopards in the Cederberg appear heavily reliant on hyrax and klipspringer in particular, which together constitutes ~61% of biomass consumed (*Martins et al., 2010*; *Müller et al., 2022a*). Despite lower relative abundances recorded for both these prey species in Piketberg (Table S2), where leopard diet composition remains unknown, the leopard population here appeared unaffected. Our findings thus suggest the persistence of leopards in this agriculturally transformed region, highlighting the adaptability, but also the potential vulnerability, of these large apex predators. Leopards in the Soutpansberg and Waterberg mountains (Limpopo Province) of South Africa have also been shown to thrive outside of protected regions (*Chase Grey, Kent & Hill, 2013*; *Swanepoel, Somers & Dalerum, 2015*), as is the case across a densely populated agricultural landscape in India (*Athreya et al., 2013*). Therefore, together with our results, it is evident that a landscape level approach is needed to ensure that the appropriate conservation policies, laws, and practices are implemented (*Athreya et al., 2013*) to ensure the safeguard of leopards throughout the entirety of their distributional range and not just in protected areas. The importance of protected regions is, however, not to be underestimated (see *Mohammadi et al., 2021*).

**Natural prey species availability**
Our camera surveys appear to have recorded almost all natural potential prey species (mammals >0.5 kg) present, as the number of species captured was very close or equal to

the total number estimated in both communities. Community composition of potential prey species did differ, but niche composition remained relatively intact across both regions, and each supports complete communities of carnivores and herbivores (Fig. 3). Generally, carnivores do not play a significant role in leopard diet in the Western Cape (*Martins et al., 2010*; *Drouilly, Nattrass & O'Riain, 2018*; *Mann et al., 2019*; *Müller et al., 2022a*), although elsewhere small carnivores are commonly killed (*Palomares & Caro, 1999*; *Hayward et al., 2006*). Herbivores unique to each community in our study (gemsbok in Cederberg; kudu in Piketberg: Table S2) were both large antelope species (>100 kg) which also generally do not constitute a major component of leopard diet in the Western Cape, except in the Little Karoo (*Martins et al., 2010*; *Drouilly, Nattrass & O'Riain, 2018*; *Mann et al., 2019*; *Müller et al., 2022a*). Here, their marked presence in leopard diet has been attributed to land-use change, characterised by an increase in game farming in recent years (*Mann et al., 2019*). It is however improbable that the very low abundances (Table S2) and limited distributions (detected at only one station each) we recorded for these large antelopes in Piketberg would effectively support its leopard population. Instead, predation on these introduced individuals would be likely to ultimately result in farmer-predator conflict and increase the level of risk that leopards are exposed to (*Constant, Bell & Hill, 2015*). Hence, the minor differences observed in potential prey species composition and richness between the Cederberg and Piketberg are unlikely to influence leopard diet substantially, and consequently population persistence, in the latter.

## Variance of main prey species

Leopards in Piketberg are more likely to rely on alternative main prey species to those primarily utilised in the Cederberg, based on observed differences in their perceived availability in each community (Table S2). By considering the factors that strongly affects habitat use of main prey species (Table 1), we broadly infer some potential underlying drivers of these differences. Due to the inability to model probabilities of space-use and detection for baboon and rhebok, we did not speculate on probable causes for differences in their availability (*i.e.*, RAIs). It should also be noted that our results for hyrax may be unintentionally skewed because of their restricted habitat (being confined to rocky outcrops; *Skinner & Chimimba, 2005*) not being accounted for in our camera set-up procedures; we advise caution in interpretation thereof.

Unsurprisingly, the relative impact on prey species by anthropogenic factors accompanying agricultural practices (*e.g.* roads, disturbances, habitations) was greater in Piketberg, but not all species were negatively influenced (Table 1). In essence, anthropogenic drivers were generally more likely to strongly influence preferred main prey species of the Cederberg (hyrax and klipspringer) in a negative manner, but alternative main prey species (duiker, grysbok, and porcupine) appeared less sensitive, some even showing a potential preference for disturbed sites. These alternative prey species are therefore likely to play a relatively larger role in leopard diet and facilitation of the leopard population in Piketberg. The greater perceived abundance of duiker, grysbok, and porcupine in the agriculturally transformed region may further be facilitated by a preference for readily available crop food resources as these species are known to frequent

the fringes of agricultural land (*Birss, Relton & Selier, 2016*; *Bragg & Child, 2016*; *Palmer et al., 2016*). Elsewhere in the Western Cape, however, it has recently been shown that duiker and grysbok both remain dependent on natural vegetation even within severely transformed landscapes (*Jansen van Vuuren, Fritz & Venter, 2022*). We therefore believe that suitable natural habitat within the mosaic Piketberg landscape plays a pivotal role for sustaining these species. Thus, we promote the maintenance of natural vegetative corridors within and between transformed lands. Remaining natural vegetation in Piketberg can generally be considered denser than across the Cederberg, thereby having the potential to further cater for grysbok and porcupine which are both known, and shown by this study, to require sufficient vegetative cover (*Bragg & Child, 2016*; *Palmer et al., 2016*). In contrast, the sparser Karoo vegetation of the Cederberg appear to be favoured by hyrax and klipspringer (*Birss et al., 2016*; *Visser & Wimberger, 2016*). These two species also appeared to be the most affected by top-down influences of predators as both species in both communities seemingly avoided areas heavily utilised by caracal and black-backed jackal, resulting in an indirect spatial overlap with leopards in the Cederberg. Interestingly, leopard RAI strongly influenced main prey species in Piketberg only, exhibiting direct overlap with grysbok. Yet again, hyrax and duiker in Piketberg appeared to be influenced in an opposing manner. In the Cederberg and elsewhere, hyrax are particularly favoured as prey by both leopard and caracal (*Hayward et al., 2006*; *Müller et al., 2022a*). It is thus also plausible that hyrax may in fact experience significant predation pressure and therefore their numbers and activity may appear relatively limited in areas shared more frequently with leopards in Piketberg (*Wittmer, Sinclair & Mclellan, 2005*).

## Livestock—an alternative food source?

Aside from alternative main prey species playing an important role, unnatural prey (*i.e.*, livestock or domestic species) might also supplement leopard diet in Piketberg. In a national park in Pakistan and a human-dominated landscape in India for example, leopards have previously been shown to be almost completely dependent on livestock and other domestic species as prey (*Shehzad et al., 2015*; *Athreya et al., 2016*). Our decision to exclude livestock and other domestic species from abundance analyses was because the vastly greater use of livestock proof fences in the Piketberg region prevented accurate and comparable detection of livestock at camera-trap stations. While fences do not prevent the movement of leopards across a landscape, the energetic costs to leopards that accompany their presence may be a driving factor for the killing of more livestock (*Wilmers et al., 2017*). *Müller et al. (2022a)* showed that 7% biomass of leopard diet in the Cederberg was comprised of livestock. Subsequently, we believe a greater proportion can be expected in Piketberg, which is a conclusion that seems to be supported by higher levels of livestock predation events reported in recent years (C Luyt—Community Outreach Officer at the Cape Leopard Trust, C Luyt, 2020, personal communications). Although not found to be strongly correlated in either region, the inverse relationship between space-use by leopards and presence of livestock in the two contrasting study regions (Table 1), is noteworthy. Leopards in the Cederberg tended to avoid areas with signs of livestock, while in Piketberg they appear to have shown a greater preference for areas with livestock. This observation

could suggest a plausible tendency for greater reliance on livestock by leopards in Piketberg. In the Cederberg, livestock roam more freely but tend to be guarded by herders, whereas in Piketberg they are generally fenced and unguarded. Unguarded, fenced livestock that are not completely predator-proofed may result in leopards being attracted to livestock as prey, especially when preferred natural prey abundances are low (*Odden, Nilsen & Linnell, 2013*; *Khorozyan et al., 2015*). This may subsequently result in an increase in illegal retaliatory killings, making the leopard population in Piketberg particularly vulnerable (*Inskip & Zimmerman, 2009*; *Soofi et al., 2022*).

Importantly, if livestock serve as regular prey, this has the potential to result in less pressure and reduced ecological regulation of natural prey species, regarded as agricultural pests (*Norton, 1980*; *Kingdon, 1982*; *Estes, 1991*; *Skinner & Chimimba, 2005*). Examples of this have been described for other apex predators like snow leopards (*Panthera uncia*, Schreber, 1775) in Nepal and dingoes (*Canis lupis dingo*, Meyer, 1793) in Australia (*Johnson & Wallach, 2016*; *Shrestha, Aihartza & Kindlmann, 2018*). Conversely, leopards that are less reliant on livestock as prey, provided sufficient preventative measures for livestock predation are employed, can partially limit population explosions of their natural prey species (*O'Bryan et al., 2018*). Effective preventative mitigation of conflict with crop pests (*i.e.*, prey species) may also cater for leopards, offering sufficient natural prey availability in the future which may minimise livestock losses (*Odden, Nilsen & Linnell, 2013*; *Khorozyan et al., 2015*). Hence, leopards and their prey species can be regarded as assets on a landscape scale, and pro-active (preventative and non-lethal) measures employed together by livestock and crop farmers in an agricultural community can thus benefit them both instead of either one indirectly inducing conflict onto the other. We therefore agree with *Du Toit, Cross & Valeix (2017)* that a shift in attitude towards asset management, rather than problem control by means of retaliatory killings, would be advantageous as the removal of conflict-prone species is likely to be counter-productive for the community (*Conradie & Piesse, 2013*; *Lennox et al., 2018*).

### Impacts on leopard behaviour

Besides potential for retaliatory killings of leopards, a further direct negative impact due to humans was reflected by the avoidance of human habitations by leopards in both study regions (Table 1; Figs. 4A and 5A). This reiterates that anthropogenic development is responsible for habitat loss (*Swanepoel et al., 2013*; *Jacobson et al., 2016*). Environmental factors that directly dictated the spatial dynamics of leopards differed between the two regions (Table 1). Greater detectability further away from water in the Cederberg, where sources are presumably more limited, can be explained since leopards are largely independent of water (*Bothma, 2005*). Leopards in this region did however show a preference for areas closer to permanent (*e.g.* rivers; Table S1) rather than seasonal (*e.g.* streams; Table S1) water sources, but this is likely to be a simple consequence of the seasonal water sources being dry and unusable at the time of our study. The lower detection probability of leopards in Sandveld vegetation and areas presumably used for grazing in Piketberg indicate that fine-scale habitat utilisation by leopards is likely to be selective and affected by agricultural activities. The influence of altitude observed in the

Cederberg is assumed to be a consequence of the slightly higher density recorded for leopards in the region during the summer months, which relates to individuals occupying smaller home ranges on the more accessible lower mountain slopes at this time of year (*Müller et al., 2022b*). Finally, the relationship we observed between leopards and meso-carnivores in the Cederberg is consistent with the conclusions of *Müller et al. (2022a)*: caracal (meso-predator) tend to avoid leopards (apex predator) in time and space. Interestingly, the availability of natural main prey species did not directly dictate spatial utilisation by leopards, which is probably because leopards (and natural prey) occurred widely across the study area. It is important to note that the impacts we observed on the behaviour of various main prey species, both environmental and anthropogenic, do have the potential to indirectly affect leopards going forward. Therefore, continuous monitoring and evaluation of the environmental and anthropogenic factors affecting the ecology of both leopards and their natural prey is of vital importance to ensure the persistence of leopards in both protected and unprotected areas.

## Study limitations

Potential criticisms of our study are that data were collected roughly 2 years apart, and that we investigated factors influencing behaviour only at a single spatial scale. We are aware that community structure could be influenced by weather on a seasonal or annual basis. The Western Cape experienced a relatively dry year (*South African Weather Service, 2021*) prior to the Cederberg survey and recorded precipitation of 212.9 mm below the annual mean (2017; Data provided by www.meteoblue.com) in the region. A precipitation anomaly of 139.9 mm below the mean (2019; Data provided by www.meteoblue.com) was also noted prior to the survey in the Piketberg region. Nonetheless, *Müller et al. (2022b)* showed that the leopard population in the Cederberg remained relatively stable during the decade prior to, and including, our study period. Currently, the same quantitative insight does not exist for the Piketberg community, highlighting a need for long-term population monitoring. Ideally, ecological modelling should aim to incorporate different orders of scale (*e.g. Pitman et al., 2017*), but this is not always practical. We concur that spatial scale must, however, be accounted for in conservation decision-making.

## CONCLUSIONS

Our findings exemplify how severely transformed, commercial agricultural regions have the potential to facilitate biological diversity—including apex predators—to (at least) the same standard as analogous protected landscapes (also see *Linell, Swenson & Andersen, 2001*). Indeed, no real differences were apparent between the potential prey species composition or richness, or the prevalence of leopards, in the agriculturally transformed Piketberg and the primarily protected Cederberg communities. Despite the preferred prey species of leopards in the Cederberg being relatively less abundant in Piketberg, the leopard population here seemingly persevere. We believe that greater utilisation of alternative main prey species to those preferred in the Cederberg, likely further subsidised by livestock, facilitates persistence of the Piketberg leopard population. Consequently, this adaptation is a probable driving factor of high levels of human-wildlife conflict. Therefore, in the context

of mixed-farming communities, we argue that a holistic multi land-use, multi-species (predator and prey), pro-active management approach, that encourages co-existence and aims to limit the cascade of ecosystem effects that could follow human-induced changes to the landscape, can benefit both livestock and crop farmers. Ultimately, such a collaborative and holistic approach can provide incentive to conserve apex predators and their prey and is therefore useful to ensure the conservation of apex predators on working lands worldwide. Furthermore, we also provide insights on the different combinations of factors influencing the spatial dynamics of leopards and their main prey species. In essence, this study can be used to inform conservation policies that aim to cater for free-roaming leopards in commercial agricultural landscapes, and act as a baseline for ecological monitoring of the Piketberg community, thereby guiding adaptive management going forward. We encourage further detailed investigation of the leopard population in Piketberg, including density, home-range, population structure, dietary and human-wildlife (both leopards and their prey) conflict analyses, to further inform local conservation management decision-making and maintain its leopard population into the future.

## ACKNOWLEDGEMENTS

We are grateful to the Cape Leopard Trust who provided support and resources to enable this research. We would like to extend our thanks to all the landowners and to CapeNature for granting access to their properties and for their collaboration. We would like to extend a special thank you to Jacobus Smit for providing accommodation and contacts within the Piketberg community. We are grateful for the field assistance provided by Chavoux Luyt, Mari-Su de Villiers, Ross de Bruin, Barbara Seele, Christiaan Lochner, Ismail Wambi, Ewan Brennan, Hannes de Kock and Grant Baker, and contributions to data processing by Mari-Su de Villiers. Thank you to Kathryn Williams for providing comments on this manuscript.

### Funding

This research was supported by ABAX Investments, Bushmans Kloof Wilderness Reserve, Ford Wildlife Foundation, Hans Hoheisen Charitable Trust, Lomas Wildlife Protection Trust, and the Rolf Stephan Nussbaum Foundation. The funders had no role in study design, data collection and analysis, decision to publish, or preparation of the manuscript.

### Grant Disclosures

The following grant information was disclosed by the authors:
ABAX Investments.
Bushmans Kloof Wilderness Reserve.
Ford Wildlife Foundation.
Hans Hoheisen Charitable Trust.

Lomas Wildlife Protection Trust.
Rolf Stephan Nussbaum Foundation.

## Competing Interests

Eugene Greyling received non-monetary support from the Cape Leopard Trust. Lana Müller was an employee of the Cape Leopard Trust and currently serves on the organisation's Scientific Advisory Board. Alison J. Leslie also serves on the Scientific Advisory Board of the Cape Leopard Trust.

## Author Contributions

- Eugene Greyling conceived and designed the experiments, performed the experiments, analyzed the data, prepared figures and/or tables, authored or reviewed drafts of the article, and approved the final draft.
- Jessica Comley analyzed the data, prepared figures and/or tables, authored or reviewed drafts of the article, and approved the final draft.
- Michael I. Cherry conceived and designed the experiments, authored or reviewed drafts of the article, and approved the final draft.
- Alison J. Leslie conceived and designed the experiments, authored or reviewed drafts of the article, and approved the final draft.
- Lana Müller conceived and designed the experiments, performed the experiments, authored or reviewed drafts of the article, and approved the final draft.

## Field Study Permissions

The following information was supplied relating to field study approvals (*i.e.*, approving body and any reference numbers):

Relevant permissions to conduct our research were granted by the Social, Behavioural and Education Research Ethics Committee at Stellenbosch University, CapeNature, and in writing by all landowners involved (Stellenbosch University, Project ID: #15315; Cape Nature, Permit number: CN44-59-12321).

## Data Availability

The raw data and code are available in the Supplemental File.

## Supplemental Information

Supplemental information for this article can be found online at http://dx.doi.org/10.7717/peerj.14575#supplemental-information.

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
