# Peer review of "Facilitation of a free-roaming apex predator in working lands: evaluating factors that influence leopard spatial dynamics and prey availability in a South African biodiversity hotspot"

_PeerJ, doi:10.7717/peerj.14575_

## Round 0.1 · original submission · Major Revisions

Thank you for submitting this study to PeerJ. I regret that I am unable to accept the manuscript for publication, at least in its present form. However, I am prepared to consider a new version that carefully takes into account the suggested edits. The reviewers liked many aspects of your study but also highlighted some important parts for revision. These need to be addressed in detail in the new version. Such a revised manuscript may be reviewed again and there is no guarantee of acceptance. Please pay close attention to the Methods section of the paper, and be very clear about the ethical approval for the research, e.g. "The authors must provide an ethics statement as part of their Methods section". Does the permit cover the ethics of the research? Please provide additional information.
See this PeerJ link for additional guidance: https://peerj.com/about/policies-and-procedures/#animal-research
When you revise the study, please prepare a detailed explanation of how you have dealt with all of the reviewers' as well as my own comments.

·

Basic reporting

This paper provides important insights into leopard ecology and its prey in Africa using diversity indices and occupancy modeling. Even though the paper has been written great, it still can be improved in terms of grammatical flaws. In general, this paper lacks a flow in its introduction. Perhaps the main flaw is that they built multiple occupancy models for different prey species without incorporating each species' unique ecology into their models. Moreover, authors should provide more citations from other leopard populations both in Africa and also even Asia as of right now to me it is a local-scale study that suits local journals.

I will point out my specific comments about the whole manuscript in detail below (Additional comments section).

Experimental design

- I’m confused about why occupancy models were created for all the species? Does data availability govern your choices or …? Please explain.

Validity of the findings

I think their findings are valid, my only concern is the occupancy models as I mentioned earlier.

I appreciate that authors uploaded all of their data and r codes.

Their conclusions are not well-stated and backed up by their data, and analyses.

Additional comments

Abstract:
In general, please restructure your abstract, and instead of starting with leopards from the very beginning, talk about the bigger picture. You should provide 1-3 lines of context for the readers.

Introduction
-Your introduction in its current form is suitable for a local journal or a cat specialist journal as you talked about leopards and Western Cape a lot. What you have to do is to talk about the bigger picture and ecology, this way you can absorb more readers from diverse backgrounds.
Line 56: I would suggest restructuring the opening as in its current form its not ideal for the first sentence of the first paragraph.
Line 67: It is too early to start talking about leopards

Line 52-55: It is too early to be that specific and talk about your study species in such a general sentence. You don’t need to talk about all four of your study species ecology instead talk about the processes (habitat suitability, anthropogenic impacts, why these species matter, …) in your introduction.

Methods:
Line 175-176: What do you mean by that, you should either cite a paper that you followed or just explain it here.
Line 176-179: There is no need for this information, you can talk about it in the discussion but not here.
Line 179-181: Please mention the home range size as well.
Line 191: For the covariates table in your supplementary, please add the citation for each of them that justifies its relevance to species ecology.
Line 225: I don’t think taxonomy would be a great equivalent for species composition, its basically a field of science dealing with naming species, but us as ecologists use species composition simply for knowing the number of species, so please remove that.
Line 290: Please correct the sign before QAIC (I assume its delta)

Results:
Line 296: You mean 10114? Please remove the space.
Line 297: Please remove the space between numbers.
Discussion
Right now you only cite papers mainly from that area or some general citations about leopards (not as diverse as it should be).
You should also discuss some papers from other parts of the world on different subspecies of leopards: Indian leopard, Persian leopard, African leopard in other countries, etc.
Also on top of that, you should see the big picture and compare your results even with different ecosystems with big cats (Jaguars in South America, Tigers in Southeast Asia, Cougar in North America?).

Figure 1: The towns/villages sign in figure 2 is confusing, it is even outside of the study area boundary, so what are you trying to show?
Figure 4: The quality on these graphs is so low and the axes are hard to read, I don’t know if the journal is going to be ok with it as well, so if possible, increase the quality on these.

·

Basic reporting

The basic reporting is professional and readable throughout. The structure of the article conforms to acceptable format and the camera trap data is converted into excel files; both the excel data and R code can be opened. Clarity can be improved in some places by using simpler phrasing if there is an alternative available (e.g., extirpation vs local extinction) or dividing up longer sentences for ease of reading (e.g., lines 85 to 90). There are some areas where text can be improved for understanding, but these are detailed below.
You give good background information on leopards and their prey in this region, and sufficient knowledge to understand potential human-animal conflict in South Africa. I would like to know further information on current conservation on private lands and how much is known about leopards on these farmlands that isn’t about conflict? Are there any incentives (e.g., tourism, grants) for farmers to facilitate leopard occupancy, or are there any current measures of protection on private land? If there aren’t any, please say so. It would also be interesting to know if there are any other comparisons of apex predators between conservation land and farmland elsewhere in the world, and how this study relates to them and this knowledge gap. Are most existing leopard studies in South Africa performed in protected areas, and this is the first to examine them outside?
Figure 1 names the green area on the map as CapeNature Reserves - could you please clarify that the Cederberg areas listed in text (lines 153-154) are known collectively as the CapeNature Reserves? If not, please clarify what the CapeNature Reserves are to those unfamiliar with South Africa.
There are a few areas where it is unclear whether leopards are being included with the result numbers and it could affect the readability of the text; could you please clarify throughout when you are talking about all medium-to-large terrestrial mammals or just prey species? For example, the phrasing in-text (lines 308-310) about Table S2 and the numbers you draw from it are a little confusing; you mention “Thirty potential natural prey species were photographed across the two regions”, but Table S2 has 30 entries including leopard. My apologies if you are including leopards preying on each other, but maybe you could mention this or change the phrasing to “Thirty terrestrial mammals…” to avoid confusion?
On line 357 could you please expand or rephrase “leopards were less likely to be detected … in areas utilised for grazing (Table 1; Figure 4G & H)”, as I first interpreted it as “leopards are less likely to be detected in areas with livestock”, which suggests a difference between the figures 4G and 4H for all the vegetation types. If there is a difference, can you please draw attention to it or use a different figure, as at first glance there is none? If “areas utilised for grazing” does not equal “livestock”, please also clarify this.
Overall, the submission is self-contained and coherent; I think that the ideas and study design is great, you’ve measured a lot of environmental and anthropogenic factors, and looked at each species in depth. Your aims are clear, to determine if there was any difference in leopard and prey populations between the two areas, and the factors that affected animal distribution.

Experimental design

I believe this is an original primary research article within the aims and scope of PeerJ. I find the study interesting and meaningful, but I would like a little more information on how your research fits with the knowledge gap about apex predators outside of conservation areas (as discussed above, e.g., are there any existing studies similar and what did they find), and if yours is the first in this knowledge gap please identify and say this.
Investigation is to a high technical standard, methods and information on site selection/factors are detailed. For more information, could you please say if there was/was not any notable large differences between the two sampling years that are not listed, e.g., weather, population boom, that could have affected differences in rate?
In the methods or results, could you please clarify that you are counting the water mongoose spoor (as observed in the initial survey but not in the camera traps) in the richness estimates, as I needed some time to figure out why the numbers on line 309 are different to the richness estimates for Piketberg and not for Cederberg.
It may be helpful to have more information on the predominant livestock in this region (e.g., are they cattle?) and if possible, the distribution of the different types of the 55 farms. It is probably not possible, but did you look at any differences in occupancy between different types of farm land use, e.g., fruit vs crop?
My main concern with the methods section is that there is no ethical statement. Even observational studies require further information about ethical approval, and since you are dealing with camera traps, this may require human ethics too. Please provide an ethics statement as part of your methods section detailing full information as to approval, granting organisation, reference number and if you followed guidelines for ethical standards during this study. As you are using camera trap footage, I would also like to know whether you sought human ethics approval, as cameras can also capture images of humans throughout, and I would also like to know how you approached this data when analysing (please see Sharma et al. (2020) Conservation and people: Towards an ethical code of conduct for the use of camera traps in wildlife research. https://doi.org/10.1002/2688-8319.12033) for more information.

Validity of the findings

This study is not implied to be replicated or derivative of existing work, and apart from not being able to record livestock, I believe the authors did a good job of recording species and extrapolating information from the camera trap records. The number of species, abundance and factors affecting mammal occupancy are largely explained well and in detail. Sometimes I was a little confused about how you phrase or conclude relationships between prey and predator distribution, as some parts in the discussion are phrased in a causative, directional way. This could be improved by re-phrasing some parts on which animals “avoid” each other (for example, the relationship between hyrax and caracal), and explaining how it is not due to other confounding factors. Overall, conclusions are well-linked to the original research aims.

---

## Round 0.2 · Minor Revisions

Thank you for carrying comprehensive revisions to the manuscript. The reviewers and I are mainly happy with the new text, but the study would still benefit from some further revisions (see reviews). Would it be possible to insert the main two or main four leopard prey species into the abstract? Also, the Discussion would read better if the wider literature was more thoroughly integrated with the existing text (especially for a journal with a broad readership). I look forward to seeing the next version.

·

Basic reporting

I think the paper has improved a lot in terms of both clarify and language.
They added more citations from diverse regions. I still encourage them to use the following citations which appears to be in line with their research:

Mohammadi, A., Lunnon, C., Moll, R. J., Tan, C. K. W., Hobeali, K., Behnoud, P., ... & Farhadinia, M. S. (2021). Contrasting responses of large carnivores to land use management across an Asian montane landscape in Iran. Biodiversity and Conservation, 30(13), 4023-4037.

Snider, M. H., Athreya, V. R., Balme, G. A., Bidner, L. R., Farhadinia, M. S., Fattebert, J., ... & Kays, R. (2021). Home range variation in leopards living across the human density gradient. Journal of Mammalogy, 102(4), 1138-1148.

Soofi, M., Qashqaei, A. T., Mousavi, M., Hadipour, E., Filla, M., Kiabi, B. H., ... & Waltert, M. (2022). Quantifying the relationship between prey density, livestock and illegal killing of leopards. Journal of Applied Ecology.

Lumetsberger, T., Ghoddousi, A., Appel, A., Khorozyan, I., Waltert, M., & Kiffner, C. (2017). Re‐evaluating models for estimating prey consumption by leopards. Journal of Zoology, 302(3), 201-210.

There are still a couple of minor issues such as : Line 97: Please remove IUCN Red List of Threatened Species as it is not a citation, just use the proper citation, or the name of the author they cite is Linnell. So I think some minor revisions are needed and after they checked the whole manuscript thoroughly then the paper should be good to be published.

Experimental design

I think now they do a better job in explaining their methodology.

Validity of the findings

no comment

·

Basic reporting

There are some new readability issues as you have added a lot of new text, but these can be fixed by a careful read through, dividing up and rearranging long sentences, and checking grammar (e.g., lines 66-71, 503, 534-539, 606-607, please don’t start sentences with “And” and such). In line 179 you change detection probability to detection behaviour – are these two equal?

The rewording of the abstract, introduction and discussion has improved the context and has provided more links with other studies outside of South Africa, as well as addressing the knowledge gap. The introduction has a better structure and now includes more varied citations. However, it would be nice later to expand on the similarities/differences of existing papers (even those with other big cats) with your results about leopard prevalence in Piketberg, and overall implications in order to gain a global perspective, expanding around line 479. I also previously asked if there were any incentives/protection measures for facilitating leopards on transformed land in South Africa; please mention even if there aren’t any.

Thank you for clarifying the cannibalism in leopards. For the differences between livestock areas and non-livestock areas (Figures 5B and 5C), I appreciate that you have split the figures up for readability, but I’m still not convinced this is clear, as 5B and 5C look very similar, which may cause confusion - is there any other way you can show this difference other than these two graphs?

Experimental design

It may be useful to have a reminder of the names of the “main” prey species in the results section (e.g. those listed in introduction at line 140). Also, please clarify how you distinguished the presence/absence of disturbance in the supplementary table (was it people presence, rubbish, footpaths etc?) as it may help us to understand results such as porcupine in Piketberg being less likely to be detected near human habitation but more likely to be in areas of human disturbance (lines 430-433).

Separately, you mention that the leopard population has already been shown to remain stable during the decade before and during the years this study was performed in Cederberg – I think it would make sense to mention this in text and give some more background to the limitations. Seeing as you think your conclusions would still hold true, you could mention this population boom in Piketberg or hypothesise how the weather differences between the two years could influence community structure?

Thank you for including an ethics statement and additional information about human subjects; you could add this in-text for extra information about how you treated confidential data.

Validity of the findings

Thank you for revising your discussion section, it is easier to read now and has a better flow. There are some parts that are not particularly clear however, as some results are phrased in a way that seems as if they have the same relationship strength. The hyrax/caracal link is still a little confusing at first as to how hyrax avoid caracal but caracal choose areas with higher hyrax activity (lines 518 to 523), so may need further rephrasing on some conclusions.

---

## Author Rebuttal · Round 0.2

PO Box 31139
Tokai, Cape Town
7966, South Africa

9th of August 2022

Dear Prof Alan McElligott

Thank you for your consideration, effort, and valuable feedback on our manuscript. Please, also extend our thanks to the reviewers for their valuable comments, advice, and suggestions. We have revised the manuscript to address the concerns raised and responded to all comments in the author response document herewith attached.

You'll be pleased to note that a clear ethics statement has been added into the manuscript. We also attached a letter from the relevant ethics committee who reviewed the initial study proposal for your reference.

We believe that the manuscript is now suitable for publication in PeerJ.

Thank you.

*Greyling*

Eugene Greyling
Student Researcher
The Cape Leopard Trust / Stellenbosch University

On behalf of all authors.

Registered Trust Number: IT 2720/2004 | PBO Number: 930 016 841 | NPO Number: 192-416
Address: P.O. Box 31139, Tokai, Cape Town, 7966
Email: contact@capeleopard.org.za | Website: www.capeleopard.org.za

Board of Trustees: Johan van der Westhuizen (Chairman); Prof William Horsnell; Dr Ian McCallum;
Jannie Nieuwoudt; India Baird; Helen Turnbull
Scientific Advisory Board: Prof William Horsnell (UCT); Dr Alison Leslie (SU); Prof Dan Parker (UMP);
Dr Jacqueline Bishop (UCT); Dr Frans Radloff (CPUT); Dr Gareth Mann (Panthera)
Dr Wendy Annecke; Dr Andrew Baxter (WESSA); Lana Müller (honorary member)

# Author response to reviewer comments

**Manuscript title:**

Facilitation of a free-roaming apex predator in working lands: Evaluating factors that influence leopard spatial dynamics and prey availability in a South African biodiversity hotspot

**Authors:**

Eugene Greyling, Jessica Comley, Michael Cherry, Alison Leslie, Lana Müller

**Corresponding Author:**

E. Greyling
* * *
*Editor comments (Alan McElligott)*

"Please pay close attention to the Methods section of the paper, and be very clear about the ethical approval for the research, e.g. "The authors must provide an ethics statement as part of their Methods section". Does the permit cover the ethics of the research? Please provide additional information."

Thank you for bringing this important uncertainty to our attention. We improved clarity surrounding ethical clearance and concerns in the manuscript by including the following statement in the methods section: "Relevant permissions to conduct our research were granted by the Social, Behavioural and Education Research Ethics Committee at Stellenbosch University, CapeNature, and in writing by all landowners involved.". Ethical clearance is generally not required in South Africa for non-invasive camera trap studies such as ours, however, permission from all landowners involved was obtained to place camera-traps on their properties. Furthermore, the proposal for this study was submitted for review to the human research ethics committee (Research Ethics Committee: Social Behavioural and Education Research) at Stellenbosch University and screened by the Departmental Ethics Screening Committee in the Department of Botany and Zoology. We have obtained a letter from this committee for your reference as to why they regarded the project as not requiring any further ethics approval after initial review. This letter has been attached to our submission. Furthermore, our study obtained a research permit from CapeNature, which is the provincial conservation regulation body and authority in the Western Cape. CapeNature takes in consideration the ethical ethos and modus operandi of activities performed by research project proposals before supplying permits.

***Reviewer 1 (Danial Nayeri)***

Thank you very much for your valuable inputs and suggestions. We hope we addressed all concerns sufficiently below. The main concern regarding our decision to construct occupancy models for all species is discussed in detail below under the heading '*Experimental design*'.

*Basic reporting*

"This paper provides important insights into leopard ecology and its prey in Africa using diversity indices and occupancy modelling. Even though the paper has been written great, it still can be improved in terms of grammatical flaws. In general, this paper lacks a flow in its introduction. Perhaps the main flaw is that they built multiple occupancy models for different prey species without incorporating each species' unique ecology into their models. Moreover, authors should provide more citations from other leopard populations both in Africa and also even Asia as of right now to me it is a local-scale study that suits local journals. I will point out my specific comments about the whole manuscript in detail below (Additional comments section)."

Thank you for your feedback regarding our Introduction. We have taken your comment into consideration and revised our Introduction in an attempt to increase general flow. We have also attempted to broaden our Introduction and have included a number of relevant citations on leopard populations from across their range in the revised manuscript.

*Experimental design*

"I'm confused about why occupancy models were created for all the species? Does data availability govern your choices or …? Please explain."

We appreciate your concern with regards to our occupancy models but are a little unsure of what you mean exactly. Yes, traditional occupancy modelling which looks at a "space/habitat occupied" parameter can only be accurately determined and labelled as such if species ecology (i.e. home range size/specialised habitat) is accounted for in experimental set-up. In the case of our study, however, we were not interested in occupancy as the "(amount of) space occupied" parameter for any of the species, but rather as "(type of) space being utilised" where "type" is determined using the various covariates we considered. We do make this clear in our manuscript in lines 326-328. Please see MacKenzie & Nichols (2004) for any further clarity surrounding interpreting occupancy as space used instead of space occupied.

*Validity of the findings*

"I think their findings are valid, my only concern is the occupancy models as I mentioned earlier. I appreciate that authors uploaded all of their data and r codes. Their conclusions are not well-stated and backed up by their data, and analyses."

Thank you for raising your concern with regards to our conclusions. Your concerns regarding our occupancy models have been addressed in the previous response. We do, however, find ourselves at a bit of a crossroad as Reviewer 2 states that "Overall, conclusions are well-linked to the original research aims". In an attempt to provide clearer conclusions that are indeed backed up by our data and analyses, we have revised our Discussion section.

*Additional comments*

Abstract:

In general, please restructure your abstract, and instead of starting with leopards from the very beginning, talk about the bigger picture. You should provide 1-3 lines of context for the readers."

Thank you for your comment. Further context has been added to the beginning of the abstract.

Introduction:

Your introduction in its current form is suitable for a local journal or a cat specialist journal as you talked about leopards and Western Cape a lot. What you have to do is to talk about the bigger picture and ecology, this way you can absorb more readers from diverse backgrounds."

Thank you and we agree with you to a certain extent. We added in additional general context and citations to our introduction, but we have also kept in information regarding the leopard sub-populations of the Western Cape.

"Line 56: I would suggest restructuring the opening as in its current form its not ideal for the first sentence of the first paragraph."

"Line 67: It is too early to start talking about leopards"

"Line 52-55: It is too early to be that specific and talk about your study species in such a general sentence. You don't need to talk about all four of your study species ecology instead talk about the processes (habitat suitability, anthropogenic impacts, why these species matter, …) in your introduction."

Thank you for your valuable comments above. We have restructured our Introduction and included additional information in an attempt to address your concerns listed above.

Methods:

"Line 175-176: What do you mean by that, you should either cite a paper that you followed or just explain it here."

Thank you for your comment. We have revised this section of the manuscript to read as follows: "Our setup procedures followed standard protocols optimized for the detection of leopards, whereby the landscape across both study regions was divided into 50 km$^2$ blocks (Figure 1), based on the minimum estimated home range size recorded for a female leopard with cubs in the Western Cape (37 km$^2$; Martins, 2010). …"

"Line 176-179: There is no need for this information, you can talk about it in the discussion but not here."

Thank you for your suggestion. We opted to remove this information.

"Line 179-181: Please mention the home range size as well."

Done. The specific home range size referred to has been included.

"Line 191: For the covariates table in your supplementary, please add the citation for each of them that justifies its relevance to species ecology."

Thank you for your insightful comment. Table S1 has been updated. Take note that specific reference is made to leopards. This is because the leopard prey species of interest must share a habitat with leopards to be considered potential prey species and therefore, we were only interested in those members of each species that co-occur in space with leopards.

"Line 225: I don't think taxonomy would be a great equivalent for species composition, its basically a field of science dealing with naming species, but us as ecologists use species composition simply for knowing the number of species, so please remove that."

Thank you for bringing this to our attention, and to avoid confusion we have removed the word "taxonomy".

"Line 290: Please correct the sign before QAIC (I assume its delta)"

Thank you, the sign has been corrected.

Results:

"Line 296: You mean 10114? Please remove the space."

Done.

"Line 297: Please remove the space between numbers."

Done.

Discussion:

"Right now you only cite papers mainly from that area or some general citations about leopards (not as diverse as it should be). You should also discuss some papers from other parts of the world on different subspecies of leopards: Indian leopard, Persian leopard, African leopard in other countries, etc. Also on top of that, you should see the big picture and compare your results even with different ecosystems with big cats (Jaguars in South America, Tigers in Southeast Asia, Cougar in North America?)."

Thank you for your advice. We have included and briefly discussed several citations that refer to the bigger picture (i.e. other apex predators and populations across the world) in the revised manuscript.

"Figure 1: The towns/villages sign in figure 2 is confusing, it is even outside of the study area boundary, so what are you trying to show?"

Thank you for your concern, however, we believe that the towns/villages sign in Figure 1 is an important aspect as it provides a clear reference to the reader (both local and foreign) as to exactly where the study took place within the Western Cape.

"Figure 4: The quality on these graphs is so low and the axes are hard to read, I don't know if the journal is going to be ok with it as well, so if possible, increase the quality on these."

Thank you for this valuable comment. We have split Figure 4 into two separate figures (Figure 4 and Figure 5) and increased the text size to ensure that quality and readability is maintained for publishing.

Thank you very much for your valuable inputs and suggestions. We hope we addressed these sufficiently below. The comments about the way in which our study fills a knowledge gap are more explicitly addressed in the revised manuscript. The concern regarding ethics is dealt with in the response to the editor, who raised the same concern, as above.

*Basic reporting*

"The basic reporting is professional and readable throughout. The structure of the article conforms to acceptable format and the camera trap data is converted into excel files; both the excel data and R code can be opened. Clarity can be improved in some places by using simpler phrasing if there is an alternative available (e.g., extirpation vs local extinction) or dividing up longer sentences for ease of reading (e.g., lines 85 to 90). There are some areas where text can be improved for understanding, but these are detailed below."

Thank you for your positive feedback and comment regarding improving clarity. We have revised our manuscript and hope that we have adequately addressed your concerns.

"You give good background information on leopards and their prey in this region, and sufficient knowledge to understand potential human-animal conflict in South Africa. I would like to know further information on current conservation on private lands and how much is known about leopards on these farmlands that isn't about conflict? Are there any incentives (e.g., tourism, grants) for farmers to facilitate leopard occupancy, or are there any current measures of protection on private land? If there aren't any, please say so. It would also be interesting to know if there are any other comparisons of apex predators between conservation land and farmland elsewhere in the world, and how this study relates to them and this knowledge gap. Are most existing leopard studies in South Africa performed in protected areas, and this is the first to examine them outside?"

Thank you for your questions. We revised our Introduction and Discussion in the hope that these have now been sufficiently addressed.

"Figure 1 names the green area on the map as CapeNature Reserves - could you please clarify that the Cederberg areas listed in text (lines 153-154) are known collectively as the CapeNature Reserves? If not, please clarify what the CapeNature Reserves are to those unfamiliar with South Africa."

Thank you for your comment. We have revised our Study area text and Figure 1's caption in an attempt to better explain what CapeNature Reserves are within the Cederberg region.

"There are a few areas where it is unclear whether leopards are being included with the result numbers and it could affect the readability of the text; could you please clarify throughout when you are talking about all medium-to-large terrestrial mammals or just prey species? For example, the phrasing in-text (lines 308-310) about Table S2 and the numbers you draw from it are a little confusing; you mention "Thirty potential natural prey species were photographed across the two regions", but Table S2 has 30 entries including leopard. My apologies if you are including leopards preying on each other, but maybe you could mention this or change the phrasing to "Thirty terrestrial mammals…" to avoid confusion?"

Thank you for your valuable comment. We reviewed the manuscript and supplementary material to ensure consistency of using the correct wording where applicable and to ensure clarity. For the purposes of this study, all medium-to-large terrestrial mammals (> 0.5 kg) recorded are considered potential leopard prey species. This includes the potential for leopards preying on each other (refer to revised manuscript for referenced examples).

"On line 357 could you please expand or rephrase "leopards were less likely to be detected … in areas utilised for grazing (Table 1; Figure 4G & H)", as I first interpreted it as "leopards are less likely to be detected in areas with livestock", which suggests a difference between the figures 4G and 4H for all the vegetation types. If there is a difference, can you please draw attention to it or use a different figure, as at first glance there is none? If "areas utilised for grazing" does not equal "livestock", please also clarify this."

Thank you for raising your uncertainty. Your interpretation is correct, and we have added in the following to ensure clarity "In Piketberg, leopards were less likely to be detected at sites characterised by Sandveld vegetation and in areas utilised for grazing, as was indicated by the presence of signs of livestock". To increase the readability of the graphs we have i) split Figure 4 into two figures, one for Cederberg and one for Piketberg, ii) standardised the Y-axis for all figures, and iii) added in the estimates for the graphs which refer to the vegetation types across the areas with and without livestock.

"Overall, the submission is self-contained and coherent; I think that the ideas and study design is great, you've measured a lot of environmental and anthropogenic factors, and looked at each species in depth. Your aims are clear, to determine if there was any difference in leopard and prey populations between the two areas, and the factors that affected animal distribution."

Thank you.

*Experimental design*

"I believe this is an original primary research article within the aims and scope of PeerJ. I find the study interesting and meaningful, but I would like a little more information on how your research fits with the knowledge gap about apex predators outside of conservation areas (as discussed above, e.g., are there any existing studies similar and what did they find), and if yours is the first in this knowledge gap please identify and say this."

Thank you. We clarified the knowledge gap in the Introduction and made comparisons between our study and others in the Discussion.

"Investigation is to a high technical standard, methods and information on site selection/factors are detailed. For more information, could you please say if there was/was not any notable large differences between the two sampling years that are not listed, e.g., weather, population boom, that could have affected differences in rate?"

Thank you for your suggestion and comment. We have expanded the potential criticism / limitations section towards the end of the Discussion to address this comment. Müller et al. (2022; https://doi.org/10.1371/journal.pone.0254507) showed that the leopard population in the Cederberg remained relatively stable during the decade prior to, and including, this study period. For Piketberg, we do not have the same information yet and long-term monitoring would provide more clarity. Hence, we made recommendations regarding the leopard

population in Piketberg in particular. A population boom in Piketberg in recent years is very plausible as inferred from an increase in reported conflict incidents and informal discussions with local residents/farmers. Nevertheless, we are confident that our conclusions hold true if this leopard population is to persist going forward.

"In the methods or results, could you please clarify that you are counting the water mongoose spoor (as observed in the initial survey but not in the camera traps) in the richness estimates, as I needed some time to figure out why the numbers on line 309 are different to the richness estimates for Piketberg and not for Cederberg."

As stated in line 359 (was line 304-305) these values of S = 27 and S = 26 are the estimated values for species richness, whereas the values in line 364 (was line 309) are the actual number of species photographed in each region. Therefore, we are not counting the water mongoose spoor for either of these values. To avoid confusion, we have removed the following phrase from the manuscript "(although tracks/spoor of water mongoose were noted during the Piketberg survey)".

"It may be helpful to have more information on the predominant livestock in this region (e.g., are they cattle?) and if possible, the distribution of the different types of the 55 farms. It is probably not possible, but did you look at any differences in occupancy between different types of farm land use, e.g., fruit vs crop?"

Thank you for your valuable comment and suggestion. We have added in that livestock farming includes animals such as cattle, horses, sheep, goats and pigs (Lines 200-202). As mentioned in Line 194, the Piketberg area is characterised by mixed agricultural practices and so we have reiterated this in line 197 by saying "covering 55 privately owned mixed agricultural farms". Unfortunately, there is no up-to-date information available that we are aware of that reflects the exact land-use make-up on a fine-scale across farms in the area. There is currently, however, a study underway which examines proportion of land-use by type on a fine scale on Piketberg farms by means of questionnaires and which will likely use this information to model leopard density across specific farmland-use types. We believe that our chosen covariates do reflect land-use to some extent as fruit orchards will not have signs of livestock in the vicinity unless livestock is also being farmed for example. The major difference in land-use we were interested in was transformed vs more natural – Piketberg vs Cederberg – as is reflected by the covariates we considered.

"My main concern with the methods section is that there is no ethical statement. Even observational studies require further information about ethical approval, and since you are dealing with camera traps, this may require human ethics too. Please provide an ethics statement as part of your methods section detailing full information as to approval, granting organisation, reference number and if you followed guidelines for ethical standards during this study. As you are using camera trap footage, I would also like to know whether you sought human ethics approval, as cameras can also capture images of humans throughout, and I would also like to know how you approached this data when analysing (please see Sharma et al. (2020) Conservation and people: Towards an ethical code of conduct for the use of camera traps in wildlife research. https://doi.org/10.1002/2688-8319.12033) for more information."

Thank you for raising these concerns. Please refer to the response to the editor, above, who raised a similar concern. Furthermore, the authors and assistants who dealt with data analyses

all signed confidentiality agreements. Any photos that contained human subjects were labelled as 'human' during data processing and not utilised in any analyses. Access to all photographs was password regulated. Only highlighted photos (fauna only) are archived and safely stored by the Cape Leopard Trust for future use in short communications, science communication, or marketing.

*Validity of the findings*

"This study is not implied to be replicated or derivative of existing work, and apart from not being able to record livestock, I believe the authors did a good job of recording species and extrapolating information from the camera trap records. The number of species, abundance and factors affecting mammal occupancy are largely explained well and in detail. Sometimes I was a little confused about how you phrase or conclude relationships between prey and predator distribution, as some parts in the discussion are phrased in a causative, directional way. This could be improved by re-phrasing some parts on which animals "avoid" each other (for example, the relationship between hyrax and caracal), and explaining how it is not due to other confounding factors. Overall, conclusions are well-linked to the original research aims."

Thank you and we sincerely appreciate the suggestions made for improvement. We have revised our Discussion and hope that we have improved clarity where necessary.

---

## Round 0.3 · accepted · Accept

Nothing further. Thanks for the comprehensive final editing.